# Elemental Composition of Commercial Herbal Tea Plants and Respective Infusions

**DOI:** 10.3390/plants11111412

**Published:** 2022-05-26

**Authors:** Jaime Fernandes, Fernando H. Reboredo, Inês Luis, Maria Manuela Silva, Maria M. Simões, Fernando C. Lidon, José C. Ramalho

**Affiliations:** 1Departamento Ciências da Terra, Faculdade de Ciências e Tecnologia, Campus da Caparica, Universidade NOVA de Lisboa, 2829-516 Caparica, Portugal; jaimefsfernandes@gmail.com (J.F.); idc.rodrigues@campus.fct.unl.pt (I.L.); mmsr@fct.unl.pt (M.M.S.); fjl@fct.unl.pt (F.C.L.); 2GeoBioTec, Departamento de Ciências da Terra, Faculdade de Ciências e Tecnologia, Campus da Caparica, Universidade NOVA de Lisboa, 2829-516 Caparica, Portugal; abreusilva.manuela@gmail.com (M.M.S.); cochichor@isa.ulisboa.pt (J.C.R.); 3ESEAG-COFAC, Avenida do Campo Grande 376, 1749-024 Lisboa, Portugal; 4Plant Stress & Biodiversity Lab, Centro de Estudos Florestais (CEF), Instituto Superior Agronomia (ISA), Universidade de Lisboa (ULisboa), Quinta do Marquês, Av. República, 2784-505 Oeiras, Portugal

**Keywords:** elemental composition, herbal tea samples, herbal infusions, inductively coupled plasma-atomic emission spectroscopy, PCA analysis, X-ray fluorescence

## Abstract

This study evaluated the elemental composition of 25 herbal tea plants commonly used in infusions by Portuguese consumers and the contribution to the elemental daily intake of some essential elements. *Hydrocotyle asiatica* (L.), *Matricaria chamomilla* (L.), and *Melissa officinalis* (L.) samples are a rich source of K with around 6.0 mg g^−1^ while the Asteraceae *Silybum marianum* (L.) and *Echinacea angustifolia* (DC.) exhibited 4.9 and 5.6 mg g^−1^ Ca, respectively. The highest concentrations of S and Zn were noted in *Hydrocotyle asiatica* (L.), while the highest concentration of Sr was found in *Cassia angustifolia* (Vahl.). In general, a large variability in the concentrations among different families and plant organs had been observed, except Cu with levels around 30 μg g^−1^. The principal component analysis (PCA) showed positive correlations between Zn and S and Sr and Ca, also revealing that *Hydrocotyle asiatica* (L.), *Echinacea angustifolia* (DC.), *Silybum marianum* (L.), and *Cassia angustifolia* (Vahl.) samples, stands out about all other samples regarding the enrichment of macro and micronutrients. The elemental solubility of macronutrients in the infusion is greater than the micronutrient solubility, despite the contribution to the recommended daily intake was weak. As a whole, *Cynara scolymus* (L.) and *Hibiscus sabdariffa* (L.) are the species with the best elemental solubilities, followed by *Hydrocotyle asiatica* (L.). No harmful elements, such as As and Pb, were observed in both the raw material and the infusions.

## 1. Introduction

Herbal tea is, after water, the beverage with the highest consumption in the world. In addition to being a simple drink, its consumption is associated with therapeutic properties of the plant’s constituents, and its therapeutic use has been used for millennia and is currently validated by medicine and pharmacology [1]. According to UN data, it is used by about 80% of the world population [2]. If there were any doubts, just consider that the market for medicinal plants is showing strong growth indicators; since 2010, the sale of infusions in Europe has increased by around 17% [3], and in the global forecast the value market will reach 5000 billion US$ in 2050 [4]. With globalization, the trade of medicinal plants increased largely, and it is well known that some particular origins are suspicious in both correct identification and its phytosanitary condition [5].

As a functional drink, tea is beneficial to human health due to their different compounds with antioxidant, anti-inflammatory, anti-bacterial, and anti-viral properties, plus protection against cardiovascular disease, hyperglycemia, metabolic disorders, and some cancers [6]. Many of these properties are attributed to secondary metabolism organic compounds such as terpenoids, phenolic compounds, and alkaloids.

Tea plants can also be seen as sources of mineral elements such as Ca, Cu, Fe, K, S, and Zn, which are important in human physiology, and some of them are often absent or in vestigial concentrations in human diets [7], leading to the implementation of different biofortification nutrient programs with success [8,9].

On the other hand, the presence of As, Cd, Ni, and Pb can be of concern due to their high toxicity, even in minimal doses, although essential elements such as Cu, Fe, and Zn can affect different biochemical pathways depending on the intracellular concentration [10,11,12]. In fact, several of the so-called natural products, whose consumption is stimulated in the West by vegan, vegetarian, and ordinary diets due to the richness in vitamins, proteins, antioxidants, and fiber [13] often contain undesirable elements [14], which is linked with improper agricultural practices related to the use of soils highly contaminated by metals [15] or the excessive use of fertilizers, herbicides, and pesticides [16].

For this reason, it is very important to know the elemental composition of the main raw material—tea leaves, fruits, cereals, algae, and horticultural products—in order to prevent harmful effects to human health [17].

Regarding the estimated daily intake (EDI), the transfer of each element from the plant matrix to the infusion must be considered. In the case of herbal teas, the factors that affect the transfer are not only the solubility of the chemical elements themselves but also how the plant material is presented (entire leaves or powdered samples), the time of extraction, and the temperature. Other factors such as geographic origin, type of soil, and anthropogenic action must be taken into account as they affect the concentration of elements in the plant [18].

Dutch and Portuguese traders were the first to introduce Chinese tea into Europe—the Portuguese shipped it from the port of Macao and the Dutch brought it to Europe via Indonesia [19]. According to a consumption study (*n* = 489) carried out in 2020 (involving mainly residents in the Oporto metropolitan area, in the North of Portugal), 76% of respondents have a high (≥5 times/week) or moderate (≤4 times/week) frequency of herbal infusion consumption, while the remaining have a consumption of ≤3 times/month [20]. The same study revealed the consumption preferences: Lemon Balm (75.9%), Chamomile (60.3%), Linden (39.1%), Peppermint (31.3%), Lemongrass (23.1%), Lemon-Thyme (16.8%), Rooibos (10.8%), and Lemon verbena (7.8%). It was also observed that the Portuguese consumer prefers herbal tea over tea itself (*Camellia sinensis* L.) [20].

The aim of the present study is to evaluate the elemental composition of 25 different plant samples with a huge diversity of geographic origin, commonly used in herbal tea preparations, presented as loosed samples and not tea bags, and available in the market stores in Portugal, by X-ray fluorescence spectroscopy (EDXRF) and simultaneously evaluate the transfer of the elements to the infusion through inductively coupled plasma-atomic emission spectrometry (ICP-AES).

## 2. Results

### 2.1. Herbal Tea Plant Characterization

All the 25 samples were purchased in supermarkets. The samples are distributed by five families as indicated: *Asteraceae* (nine samples), *Lamiaceae* (six samples), *Malvaceae* (four samples), *Apiaceae* (three samples), and *Fabaceae* (three samples)—Table 1. According to the label the origin of the samples is the following: from Portugal, 12; India, 3; Spain, 2; Egypt, 2; Bulgaria, China, Poland, 1 sample; and from the European Union 2 samples.

### 2.2. Elemental Composition

Within the Apiaceae, the Got sample (*Hydrocotyle asiatica*) always exhibited the highest concentrations of the macronutrients and micronutrients (except Cu) while the lowest concentrations were generally observed in *Foeniculum vulgare* (Fen). Regarding the family Asteraceae, a huge variability was detected within the species since the highest concentrations of Ca, K, and S were randomly observed in the studied species (Table 2). Conversely, within the Fabaceae the highest concentrations of all the macronutrients were always found in *Cassia angustifolia* (Sen) and the lowest ones in the *Trigonella foenum-geaecum* (Fgr), generally (Table 2).

In the Lamiaceae it can be emphasized the highest concentrations of Ca and S in *Lamium album* (Wne) with 4.4% and 0.6%, respectively, while in the Malvaceae the *Malva sylvestris* (CMa) contained the highest values of Ca, K, and S (Table 2). In the Asteraceae family, the highest concentration of Fe (1019 µg g^−1^) was observed in *Arctium lappa* (Gbu)—this element is dominant in the majority of the samples, except *Equinacea angustifolia* (Ech) and *Achillea millefolium* (Yar)—Table 2.

Regarding Sr, the levels range between 17.7 and 78.5 µg g^−1^. In a similar manner, the concentrations of Zn exhibit a large variation within the species, while for Cu the overwhelming average values are around 30.0 µg g^−1^ except the levels observed in *Matricaria chamomilla* (Cha) and *Achillea millefolium* (Yar), which are 42.5 and 53.9 µg g^−1^, respectively (Table 2).

Within the Fabaceae, the sample of *Pterospartum tridentatum* (Car) shows the maximum values of all micronutrients except Sr. The maximum detected for this element was 130 µg g^−1^ in Sen samples, a concentration at least ten times higher than the levels observed in the other samples of the family (Table 2).

In the Lamiaceae and Malvaceae, there is great variability in the samples analyzed. In the latter family, however, *Malva sylvestris* (CMa) can be highlighted as having the highest concentration of all the macronutrients under study, plus the highest Zn level (Table 2).

The data obtained allowed to draw average profiles of each of the families—K is the predominant element in the five families, while Ca ranks second. When considering the micronutrients, Fe ranks first followed by Zn and/or Sr. The element with the lowest concentration in all the families is Cu (Table 2).

### 2.3. Infusion Elemental Composition

The analysis performed by ICP-AES allowed us to obtain the values of diffusion of the chemical elements from plant material to water. Regarding macronutrients, the one that presents the greatest diffusion is S, which had a maximum solubility in Fen (96.9%) and Art (88.5%). Furthermore, in the overall samples only 4 out of 25 had solubilities <30%, while in the case of K, 4 out of 25 had solubilities lower than 20%. The solubility of both elements exhibits a wide range between 14.9–96.9% for S and 14.6–85–8% for K (Table 2).

In the case of Ca, the solubility ranges between 5.42% and 52.2% with the highest levels observed in Ros (52.2%), Art/Ech (45.4%) and Fgr (40.8%). Within the micronutrients, Sr presented the highest diffusion rates: 9 out of 25 samples had solubilities >40%, and the highest value was observed in Art with 89.7%. Regarding the remaining elements (Cu, Fe, and Zn) the solubility in the brew is generally poor, decreasing in the following order: Zn > Cu > Fe. In the case of Fe, the solubility ranges between 0.40% (MMa) and 13% (Ros) while for Cu it ranges between 2.15% (Lin) and 16.1% (Gbu). As a whole, Art (Asteraceae) and Ros (Malvaceae) are the species with the best elemental solubilities.

### 2.4. Principal Component Analysis

In order to integrate the analysis of the elements in the different samples, a principal component analysis (PCA) was carried out. The analysis was applied to normalized data and was performed on the attributes (chemical elements): Ca, K, S, Sr, Zn, Cu, Fe.

The first three main components account for 64%: 31% for the first main component (PC1), 18% for the second (PC2), 15% for the third (PC3). Only these components were considered significant since only these components showed a correlation with absolute value >7 with one or more original variables. In order to understand the relative importance of each attribute in relation to each of the first three main components, the correlation coefficients between the attributes (original parameters) and the main components were determined (Table 3). The PC1 was explained by the positive correlations among S, K, and Zn while the PC2 by Sr and Ca also positively correlated (Table 4).

The measurement was performed using the elemental contents as variables, the Euclidean distance as the similarity measurement, and the Ward’s method as the amalgamation rule. These clusters allow us to confirm that Got’s sample stands out above all other samples, appearing in an isolated cluster. We can see that samples such as Ech, Mth, and Sen stand out within their clusters, despite Mth and Ech belong the Asteraceae family while Sen belongs to the Fabaceae. Another cluster includes three species from the Lamiaceae (Wne, Thy, and Lba) and one from Malvaceae (Cma) and Asteraceae (Dan). Gbu and Cha are within the same cluster and belong to the same family (Asteraceae) while the remaining species are in a cluster apart (Figure 1).

## 3. Discussion

Calcium and K concentrations are by far the elements with the highest average values. In peppermint (Pep) and chamomile (Cha), Kara [22] observed 1.17 and 0.69 mg g^−1^ Ca, respectively, while for K the concentrations were more alike—1.72 for Pep and 1.84 mg g^−1^ for Cha. In our case, the minimum Ca and K levels in the same species are 1.1 for Ca (Cha) and 3.3 for K (Pep). Regarding linden (Lin) and senna (Sen) 1.41 and 2.69 mg g^−1^ Ca were measured [22], which agree with our average values of 1.4 and 2.5 mg g^−1^ Ca in the same species. If K is considered, the results were quite divergent—the levels observed in the current study were 2.2 times higher and 2.1 times lower than the average values found in linden and senna, respectively [22].

Beyond the huge variability regarding the elemental levels within the same species, we must stress that in our case we have label information about the origin of the samples i.e., from Poland (Pep), Egypt (Cha), Portugal (Lin), and India (Sen) despite the fact that they were purchased in different supermarkets or herbal stores in Portugal, while the study of Kara [22] only refers that samples were bought in a supermarket in Turkey without identifying the origin.

The Cu and Fe levels of chamomile flower heads, determined by inductively coupled plasma optical emission spectrometry (ICP-OES) were 29.1 μg g^−1^ Cu and 467 μg g^−1^ Fe, respectively [23], contrasting with our average values which were 42.5 and 598 μg g^−1^, respectively. The analysis of *Echinacea purpurea* with origin from Poland by ICP-MS revealed a very high concentration of Ca (13.7 mg g^−1^), while for Cu, Fe, and Zn, 14.0 μg g^−1^, 199 μg g^−1^ and 23.9 μg g^−1^ were observed, respectively [24]. Our results from *Echinacea angustifolia* with Portuguese origin showed for Ca 5.6 mg g^−1^, while for Cu, Fe, and Zn, 27.8 μg g^−1^, 60.2 μg g^−1^, and 19.1 μg g^−1^, respectively.

When comparing the average values of Ca, Cu, and Fe of *Achillea millefolium* (Yar), *Cassia angustifolia* (Sen), and *Melissa officinalis* (Lba) with the same species originated from Turkey it emerges that great differences were generally observed especially in Cu. The Cu levels detected in the mentioned species do not reach 1 μg g^−1^ [25], while our levels range between 26.5 μg g^−1^ (Sen) and 53.9 μg g^−1^ (Yar). Regarding Fe, the levels are very similar to ours except Yar, while our Ca levels were clearly higher in the case of Sen (2.5 mg g^−1^) and Lba (2.7 mg g^−1^) versus 1.12 mg g^−1^ in the case of Sen and 0.95 mg g^−1^ for Lba [25]. Only in Yar the Ca levels were very close.

The huge variability in the elemental composition between identical species is not explained only by the differences in the origin of the samples and analytical methodology used. Regarding the origin, soil conditions, rainfall, altitude, the age of the plant when harvested, and, in case of infusions, the amount of plant material relative to water, infusion time and temperature, are factors that influence the elemental composition in the raw material initially and in the brew thereafter [26], beyond processing methods suffered by raw material (drying, entire leaves, or powdered samples), plus packaging.

The presence of Sr in plant material is recognized by several authors despite this element is not essential for both plant and human life. In infusions, Sr might well interfere with the uptake of Ca at intestinal epithelium level, due to the chemical behavior similar to Ca. Moreover, when entering into the human body, Sr is deposited mainly in bone precisely due to its high content in Ca, approximately 40% [27]. Furthermore, the PCA analysis indicated they are correlated positively.

In PC1, the positive correlation within S and Zn is well recognized since Zn−S bonds serve as structural braces in protein domains that participate in several molecule interactions [28]. In the same context, Zn application improved K uptake, under normal as well as saline conditions, by basil (*Ocimum basilicum* L.) [29], while K was the unique element amongst those measured, which showed a positive correlation with S content in shoots of the Chinese cabbage, highlighting its role as a counter cation for sulfate during xylem loading and vacuolar storage in leaves [30]. In PC2, the positive correlations between Ca and Sr are explained by their similar chemical and physical properties as a result of the similarity of the outer electron structure of these elements [27].

### Elemental Phytotherapeutic Potential

The herbal samples analyzed in this work all exhibited therapeutic properties. Plants such as *Hydrocotyle asiatica* (Got), *Melissa officinalis* (Lba), and *Matricaria chamomilla* (Cha) contained a concentration of 6%, 5.8%, and 5.6% K, respectively. Potassium intake has been reported to be potentially beneficial for the prevention and control of blood pressure and stroke [31].

The solubility of K in the brew ranges between 14.6% (Car) and 85.8% (Wne). For example, in Cha (*M. chamomilla*) the infusion contains 311 µg mL^−1^ K, compared with the values of 456 µg mL^−1^ and 278 µg mL^−1^ observed in the infusion and obtained by total reflection X-ray fluorescence analysis [32]. It must be stressed, however, that the highest value of Winkler et al. [32] was from *Matricaria recutita* (synonym: *M. chamomilla*) with origin in Austria and the second one from Egypt, although purchased in Austria. Our Cha sample has also origin in Egypt, thus the variability within samples from some plant species, similar plant parts (flower) and country might well be related with the factors previously discussed [26].

Also linked to the formation and maintenance of bone health, Ca is an important nutrient also for the regulation of muscle function and the nervous system [33]. In the analyzed samples, the highest Ca concentrations were found in Echinacea (5.6%), milk thistle (4.9%), and white nettle (4.4%), with solubilities of 45.4% (509 µg mL^−1^), 34.4% (337 µg mL^−1^), and 6.60% (58.1 µg mL^−1^), respectively.

In the case of *Malva sylvestris* and *Mentha piperita,* the migration of Ca into the brew occurs at levels of 261 and 101 µg mL^−1^, respectively [32], while in the current study very close values were measured—261 µg mL^−1^ and 116 µg mL^−1^ Ca, respectively. Our samples come from Portugal and Poland, respectively, while those from [32] come from Austria.

The highest S concentrations were found in the samples of gotu kola (9475 µg g^−1^), common mallow (6630 µg g^−1^), and white nettle (5969 µg g^−1^), corresponding to the following solubilities: 50.6%, 59.8% and 32.5%, respectively. Sulfur is essential for the formation of connective tissue (disulfide bridge formation), formation of sulfur amino acids, and antioxidants such as glutathione [33,34]. Furthermore, the solubility of S in the studied species ranges between 14.9% and 96.9%. Similarly, the Sr solubility presents a huge variability ranging between 8.47% (MMa) and 89.7% (Art).

The levels of Cu. Fe, Sr, and Zn measured by EDXRF in *Taraxacum officinale*, *Mentha piperita,* and *Matricaria chamomilla* were very close to ours except the Sr levels [35]. However, the solubility of the same elements in infusions determined by ICP-AES when compared with similar results from the current study were, in the overwhelming majority of the cases, much higher, which is probably related to the large time of infusion used and the weight of the sample—120 min and 5 g [35]—compared with our infusion time, 5 min and only 1 g.

The lowest degrees of elemental solubility in infusions were noted in Fe, Cu, and Zn. In the case of Fe, 10 out of 23 analysis revealed percentages of solubility <1% while only 3 exhibited values >5%. Regarding Cu, the solubility ranges between 2.15% (Lin) and 16.1% (Gbu) and only four of them presented >10%, thus indicating that the contribution of both elements to the daily intake is scarce, which agrees with the data of Suliburska and Kaczmarek [36], who concluded that the herbal infusions of chamomile (flowers), mint (leaves), St John’s wort (flowers and leaves), sage (leaves), and nettle (leaves) are not important in human nutrition as sources of Ca, Mg, Fe, Zn, and Cu

Fe, Cu, and Zn are cofactors in processes of protein synthesis, thus with a fundamental role in the functioning of many organs and systems in human physiology, beyond a multitude of functions with emphasis in heme synthesis in the case of Fe, neurotransmission in the case of Cu, and finally DNA synthesis and immune response in the case of Zn [37].

The highest solubility of Zn was found in *Hibiscus sabdariffa* (Ros) with 72.4% while the lowest one was observed in *Althaea officinalis* (MMa) with 7.16%. In general, the degree of solubility of this element is much higher than the values for both Cu and Fe.

A similar conclusion was noted by Winkler et al. [32], who observed very low levels of Cu, Fe, and Zn in chamomile infusions. For example, the Cu levels range between 0.0679 and 0.112 µg mL^−1^, whereas the Fe and Zn levels ranged between 0.081–0.105 µg mL^−1^ and 0.308–1.051 µg mL^−1^, respectively. In the same context, the infusions in fennel with origin in Latvia had 0.0833 µg mL^−1^ Cu while those with origin in Austria had 0.0188 and 0.1163 µg mL^−1^ [32]. Regarding Zn, the concentrations were 0.877 µg mL^−1^ (Latvia) and 0.020 and 0.219 µg mL^−1^ (Austria).

Other authors, however, point out too many higher values in the infusions [35,36], which is probably related to the grinding process, the amount of raw material, the extraction time, the method used in the analysis beyond the different origin of the raw material, and development state of the various plant organs used in infusions.

Some authors noted that doubling the time of infusion (from 10 to 20 min) and maintaining the same amount of raw material (2 g of chamomile flowers) increases the extractability of Ca from 19.2% to 23.1% of the total, while for K this percentage increases from 65.2%to 73.4% [38]. The same authors concluded that using 1 g and 10 min only, the levels obtained for the same elements were very similar to those referred to the same extraction time [38]. From the analysis of our data, it seems that the contribution of tea drink to the daily intake of the elements under study is scarce despite the phytochemical profile of different teas with relative antioxidant and anti-inflammatory capacities has been clearly shown [39].

In a small glimpse, the estimated daily intake (EDI) calculated as follows: EDI  =  C  ×  IR/BW, where C is the concentration of a certain element (mg kg^−1^) in tea infusion, IR is the average daily consumption (g per day), and BW the body weight (kg)—we consider 65 kg as an average weight of an adult male [17]. In Portugal, the annual per capita tea consumption in 2016 was 60 g, which is equivalent to a 0.164 g consumption per day (https://www.statista.com/statistics/507950/global-per-capita-tea-consumption-by-country/, accessed on 28 April 2022) we can realize that for the 13 highest extraction rates, we obtained the following data (Table 5).

The EDI values obtained are too low when compared with the recommended daily intake levels adopted by EFSA for adults [40]. A possible way to increase the elemental extraction from the raw material is to increase the time of infusion and increase the amount of raw material used, or eventually proceed with a decoction process in which herbals are boiled during a few minutes. However, this is not a common procedure applied in households in Portugal.

It must be also emphasized that the nutrient absorption at the intestinal epithelium level reduces the load entering the blood stream even more. For example, Zn and Ca are affected by several factors such as the presence of phytates, and oxalic acid in the case of Ca. Nevertheless, the amount of protein in a meal has a positive effect on Zn absorption as well as histidine, methionine, and other low-molecular-weight ions, such as citrate [41], which may counterbalance the negative effects.

Despite the fact that these samples are included in the preferences of the Portuguese consumers, a much larger screening along with a large period is warranted in order to guarantee that habitual exporter firms maintain the quality indispensable to the safety of the public health, since national control processes and legislation are lax.

## 4. Materials and Methods

### 4.1. Elemental Determination in Plant Samples

As previously stated, samples were purchased in the market stores in Portugal. For determination of the elemental composition, herbal plants were removed from the sealed boxes (3 batches per species) and were then dried at 60 °C until a constant weight was reached. Thereafter, plant samples were reduced to powder after crushing in an agate mortar and stored. All the analysis was made using sample cups provided by the manufacturer filled with powdered plant material (on average 2.0 g) coated with a thin film of prolene.

The determination of micro and macronutrients in plant samples was performed in triplicate by using an X-ray Analyser (Thermo Scientific, Niton model XL3t 950 He GOLDD+, Waltham, MA, USA) in accordance with Environmental Protection Agency (EPA) method 6200 [42]. Detection limits using the optimum mining mode for a period of 120 s under high purity helium (He) were: Ca = 65 µg g^−1^, Cu = 12 µg g^−1^, Fe = 25 µg g^−1^, K = 200 µg g^−1^, S = 90 µg g^−1^, Sr = 8 µg g^−1^, and Zn = 6 µg g^−1^. Plant reference materials (Orchard Leaves (SRM 1571) were run before the beginning of analyses and after every five samples (Table 6).

### 4.2. Elemental Determination in Plant Infusion

Herbal infusions were prepared by adding 50 mL of boiling double DI-water to 1.0 g of tea leaves in a 50 mL conical flask. The tea infusion was mixed using a glass rod to ensure adequate wetting, then covered and allowed to boil for 5 min based on tea industry’s recommended brew time. The solution was filtered through a Whatman N° 40 filter, cooled, and diluted with double DI-water to 50 mL [43].

The determination of Ca, Cu, Fe, K, P, S, Sr, and Zn in infusions was carried out by inductively coupled plasma atomic emission spectrometry, at spectral lines λ = 422.6 nm for Ca, λ = 224.7 nm for Cu, λ = 259.9 nm for Fe, λ = 766.4 nm for K, λ = 214.9 nm for P λ = 180.6 nm for S, λ = 407.7 nm for Sr, and λ = 213.8 nm for Zn.

The analysis was carried out on a Horiba Jobin Yvon ULTIMA sequential ICP, using the Horiba Jobin Yvon ICP Analyst 5.4 software. A monochromator with a Czerny Turner spectrometer was used. The gas used was Argon. Instrument configuration and general experimental conditions are summarized in Table 7.

All the analyses were performed in triplicate and the results expressed in mg L^−1^. The detection limits of the apparatus are the following: Ca—7 µg L^−1^, Cu—4 µg L^−1^, Fe—0.3 µg L^−1^, K—20 µg L^−1^, P—13 µg L^−1^, S—14 µg L^−1^, Sr—32 µg L^−1^, and Zn—0.3 µg L^−1^. The degree of solubility of each element was calculated according to the method described by Xie et al. [44] where the solubility (expressed in percentage) of an element in a tea infusion was defined as the ratio of the mass of the element in 50 mL of the infusion (drawn from 1 g tea leaves) to the mass of the element in 1 g of tea leaves.

### 4.3. Principal Component Analysis

Principal component analysis is a dimension-reduction tool that can be used to reduce a large set of variables to a small set that still contains most of the information from the large set. This tool was applied to observe any possible clusters within the different families and species under study, with the data analysis software system, STATISTICA—version 12.

### 4.4. Statistical Analysis and Control Assurance

Statistical analysis was performed with the SPSS Statistics 18 program, through an analysis of variance (ANOVA) and the F-test. A value of *p* ≤ 0.05 was considered to be significant. All analyses were made in triplicate. Analytical accuracy was verified using replicate determinations and standard reference materials, as referred above.

## 5. Conclusions

Some families are particularly enriched in some elements such as the Apiaceae, and within this family the Got sample (*Hydrocotyle asiatica* also known as *Centella asiatica*) always exhibited the highest macronutrients concentrations. Regarding micronutrients, species from Apiaceae are a good source of Fe and Zn and even Cu, even though the average Cu values within the families were close. PCA analysis confirms that Got’s sample stands out about all other samples constituting a unique cluster. The solubility of micronutrients in the infusion is very poor, with emphasis on Fe. Regarding the macronutrients (Ca, K, and S) the highest solubilities were observed in S with 96.9% (Fen), Art (88.5%), and Rse (83.6%). In the case of Ca and K, the maximum solubility level in the infusion reaches 52.2% and 85.8%, respectively, although occurring a great variability in both cases. The subsequent estimated daily intake values, which were very low, point to a negligible contribution of the tea brew to the human daily requirements regarding essential micro and macronutrients, which somewhat agrees with the data from Portuguese annual tea consumption per capita, which is also very low. No undesirable elements (As and Pb for example), were detected in the studied samples.

## Figures and Tables

**Figure 1 plants-11-01412-f001:**
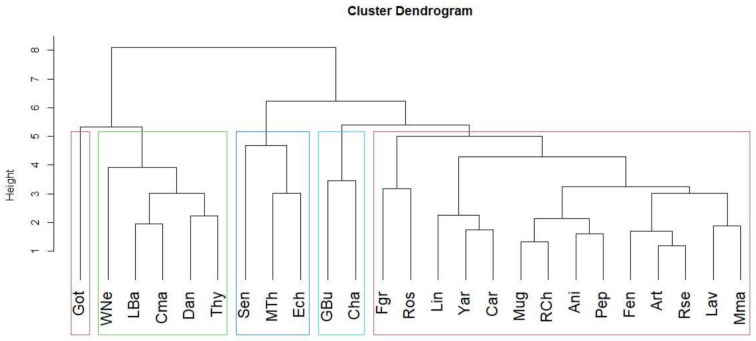
Dendrogram, obtained by the Ward.D2 method [21], from 25 plant samples, showing 5 clusters.

**Table 1 plants-11-01412-t001:** Identification of samples.

Family	Code	Common Name	Scientific Name	Plant Organ	Origin
Apiaceae	Got	Gotu Kola	*Hydrocotyle asiatica* (L.)	Aerial parts	India
	Ani	Anise	*Pimpinella anisum* (L.)	Seeds	Spain
	Fen	Fennel	*Foeniculum vulgare* Mill.	Aerial parts	Portugal
Asteraceae	Art	Artichoke	*Cynara scolymus* (L.)	Leaf	Morocco
	Mug	Mugwort	*Artemisia vulgaris* (L.)	Leaf	Portugal
	Gbu	Greater budock	*Arctium lappa* (L.)	Root	China
	Cha	Chamomile	*Matricaria chamomilla* (L.)	Flower	Egypt
	Mth	Milk thistle	*Silybum marianum* (L.) Gaertn.	Aerial parts	Portugal
	Dan	Dandelion	*Taraxacum officinale* Weber.	Aerial parts	Portugal
	Ech	Echinacea	*Echinacea angustifolia* (DC.)	Aerial parts	Portugal
	RCh	Roman Chamomile	*Anthemis nobilis* (L.)	Flower	European Union
	Yar	Yarrow	*Achillea millefolium* (L.)	Flower	Portugal
Fabaceae	Car	Carqueja	*Pterospartum tridentatum* (L.) Willk.	Flower	Portugal
	Fgr	Fenugreek	*Trigonella foenum-graecum* (L.)	Seeds	India
	Sen	Senna	*Cassia angustifolia* Vahl.	Follicle	India
Lamiaceae	Rse	Rosemary	*Rosmarinus officinallis* L.)	Leaf	European Union
	Lav	Lavender	*Lavandula spica* (L.)	Flower	Spain
	Lba	Lemon balm	*Melissa officinalis* (L.)	Leaf	Portugal
	Pep	Peppermint	*Mentha x piperita* (L.	Leaf	Poland
	Thy	Thyme	*Thymus vulgaris* (L.)	Leaf	Portugal
	Wne	White nettle	*Lamium album* (L.)	Aerial parts	Portugal
Malvaceae	MMa	Marsh mallow	*Althaea officinalis* (L.)	Root	Bulgaria
	Ros	Roselle	*Hibiscus sabdariffa* (L.)	Flower	Egypt
	CMa	Common mallow	*Malva sylvestris* (L.)	Leaf	Portugal
	Lin	Linden	*Tilia* sp. (L.)	Inflorescence	Portugal

Plant parts according to the label indication.

**Table 2 plants-11-01412-t002:** Average levels of Ca, K, S, Sr, Zn, Cu, and Fe in the different plant samples.

F	Sample	Ca	K	S	Sr	Zn	Cu	Fe
Apiaceae	Got	2.5 ± 0.04 a	6.0 ± 0.04 a	9475 ± 142 a	47.7 ± 1.5 a	115 ± 6 a	24.7 ± 8.0 b	728 ± 29.1 a
		(115 ± 1.6) 23.0%	(226 ± 2.23) 18.8%	(95.8 ± 3.94) 50.6%	(0.57 ± 0.07) 59.7%	(0.48 ± 0.08) 20.8%	(0.04 ± 0.003) 8.10%	(0.28 ± 0.02) 1.92%
	Ani	1.6 ± 0.03 b	3.5 ± 0.03 b	2349 ± 70 b	48.4 ± 1.5 a	53.2 ± 4.6 b	21.1 ± 7.3 b	242 ± 19.5 b
		(67.0 ± 1.24). 20.9%	(210 ± 2.1) 30.0%	(23.0 ± 0.96) 48.9%	(0.27 ± 0.013) 27.9%	(0.16 ± 0.012) 15.0%	(0.04 ± 0.003) 9.40%	(0.16 ± 0.014) 3.30%
	Fen	2.4 ± 0.04 a	2.6 ± 0.03 c	1496 ± 80 c	15.0 ± 1.1 b	21.3 ± 4.5 c	38.1 ± 9.5 a	BDL
		(185 ± 2.13) 38.5%	(217 ± 2.15) 41.7%	(29.0 ± 1.98) 96.9%	(0.11 ± 0.009) 36.6%	(0.21 ± 0.03) 49.3%	(0.05 ± 0.006) 6.50%	
Asteraceae	Art	2.2 ± 0.03 d	1.5 ± 0.02 f	1582 ± 66 e	36.8 ± 1.4 b	31.3 ± 4.2 e	30.5 ± 7.9 c	78.3 ± 17.3 g
		(200 ± 2.25) 45.4%	(189 ± 1.94) 63.0%	(28.0 ± 2.68) 88.5%	(0.66 ± 0.07) 89.7%	(0.06 ± 0.005) 9.60%	(0.03 ± 0.002) 4.90%	(0.14 ± 0.02) 8.90%
	Mug	3.2 ± 0.04 c	3.8 ± 0.03 b	1954 ± 81 d	17.7 ± 1.2 f	36.5 ± 4.9 d	27.8 ± 9.2 c	176 ± 21 e
		(70.7 ± 1.27) 11.0%	(205 ± 2.06) 27.0%	(12.6 ± 0.53) 32.2%	(0.07 ± 0.004) 19.8%	(0.12 ± 0.009) 16.4%	(0.04 ± 0.002) 7.20%	(0.04 ± 0.006) 1.13%
	Gbu	0.6 ± 0.02 g	1.9 ± 0.02 e	2377 ± 62 c	27.6 ± 1.1 d	37.8 ± 3.7 d	31.1 ± 6.4 c	1019 ± 27 a
		(15.0 ± 0.84) 12.5%	(171 ± 1.80) 45.0%	(24.2 ± 1.01) 50.9%	(0.10 ± 0.006) 18.1%	(0.08 ± 0.007) 10.6%	(0.10 ± 0.001) 16.1%	(0.12 ± 0.03) 0.59%
	Cha	1.1 ± 0.03 f	5.6 ± 0.04 a	2645 ± 97 b	36.8 ± 1.5 b	51.7 ± 5.3 c	42.5 ± 9.4 b	598 ± 29 b
		(31.4 ± 0.97) 14.3%	(311 ± 2.89) 27.8%	(34.4 ± 1.43) 65.0%	(0.16 ± 0.010) 21.7%	(0.24 ± 0.04) 23.2%	(0.04 ± 0.003) 4.70%	(0.08 ± 0.013) 0.67%
	Mth	4.9 ± 0.06 b	3.7 ± 0.04 b	1222 ± 98 f	78.5 ± 2.0 a	64.8 ± 5.4 b	27.8 ± 8.7 c	389 ± 24 d
		(337 ± 3.28) 34.4%	(370 ± 3.31) 50.0%	(17.1 ± 0.72) 70.0%	(0.85 ± 0.11) 54.1%	(0.24 ± 0.015) 18.5%	(0.08 ± 0.010) 14.4%	(0.06 ± 0.007) 0.77%
	Dan	2.3 ± 0.04 d	3.8 ± 0.04 b	2860 ± 86 a	24.1 ± 1.2 e	75.0 ± 5.5 a	23.0 ± 8.1 c	526 ± 26 c
		(165 ± 1.98) 35.9%	(353 ± 3.22) 46.4%	(35.8 ± 1.48) 62.6%	(0.15 ± 0.009) 31.1%	(0.32 ± 0.04) 21.3%	(0.06 ± 0.007) 13.0%	(0.10 ± 0.002) 0.95%
	Ech	5.6 ± 0.05 a	3.0 ± 0.03 d	1830 ± 76 d	32.6 ± 1.3 c	19.1 ± 4.0 f	27.8 ± 9.4 c	60.2 ± 17.3 g
		(509 ± 4.59) 45.4%	(245 ± 2.38) 40.8%	(16.9 ± 0.71) 46.2%	(0.39 ± 0.027) 59.8%	(0.14 ± 0.012) 36.6%	(0.02 ± 0.003) 3.60%	(0.03 ± 0.005) 2.49%
	RCh	1.8 ± 0.03 e	3.4 ± 0.03 c	1033 ± 73 g	31.1 ± 1.5 c	31.1 ± 5.1 e	25.9 ± 10.1 c	122 ± 21 f
		(19.5 ± 0.88) 5.42%	(192 ± 1.96) 28.2%	(11.3 ± 0.48) 54.7%	(0.07 ± 0.006) 11.2%	(0.18 ± 0.016) 28.9%	(0.06 ± 0.009) 11.6%	(0.03 ± 0.006) 1.23%
	Yar	1.2 ± 0.03 f	3.5 ± 0.04 c	1399 ± 104 f	17.9 ± 1.4 f	56.4 ± 6.8 b	53.9 ± 12.7 a	43 ± 12 h
		(35.0 ± 1.00) 14.6%	(258 ± 2.48) 36.8%	(17.8 ± 0.74) 63.6%	(0.07 ± 0.008) 19.6%	(0.28 ± 0.024) 24.8%	(0.08 ± 0.012) 7.42%	(0.03 ± 0.004) 3.49%
Fabaceae	Car	0.5 ± 0.02 b	2.8 ± 0.03 b	1557 ± 88 c	11.9 ± 1.0 c	49.9 ± 5.0 a	38.2 ± 8.5 a	120 ± 19 a
		(9.6 ± 0.72) 9.60%	(81.9 ± 1.10) 14.6%	(8.7 ± 0.37) 27.9%	(0.02 ± 0.002) 8.40%	(0.22 ± 0.017) 22.0%	(0.02 ± 0.002) 2.62%	(0.08 ± 0.015) 3.33%
	Fgr	0.2 ± 0.01 c	1.3 ± 0.02 c	2616 ± 70 b	13.6 ± 0.8 b	46.5 ± 4.0 a	16.8 ± 6.2 b	BDL
		(16.3 ± 0.85) 40.8%	(56.5 ± 0.90) 21.7%	(8.9 ± 0.17) 17.0%	BDL	(0.08 ± 0.005) 8.60%	BDL	----
	Sen	2.5 ± 0.04 a	4.6 ± 0.04 a	4724 ± 135 a	130 ± 3 a	31.0 ± 4.9 b	35.8 ± 9.6 a	96.7 ± 19.6 b
		(46.4 ± 1.08) 9.28%	(180 ± 1.87) 19.5%	(14.1 ± 0.59) 14.9%	(1.1 ± 0.076) 42.3%	(0.16 ± 0.012) 25.8%	(0.02 ± 0.003) 2.79%	(0.08 ± 0.011) 4.14%
Lamiaceae	Rse	1.8 ± 0.03 f	1.6 ± 0.02 e	968 ± 66 e	29.0 ± 1.3 c	25.8 ± 4.5 e	34.4 ± 9.1 a	317 ± 24 a
		(97.7 ± 1.47) 27.1%	(142 ± 1.57) 44.4%	(16.2 ± 0.88) 83.6%	(0.25 ± 0.033) 43.1%	(0.14 ± 0.01) 27.1%	(0.04 ± 0.005) 5.81%	(0.03 ± 0.004) 0.47%
	Lav	2.2 ± 0.03 e	2.6 ± 0.03 d	3097 ± 95 d	75.8 ± 2.1 a	28.0 ± 4.9 e	40.2 ± 10.0 a	198 ± 22 c
		(66.4 ± 1.23) 15.1%	(214 ± 2.13) 41.1%	(38.8 ± 1.61) 62.6%	(0.56 ± 0.061) 36.9%	(0.18 ± 0.03) 32.1%	(0.04 ± 0.002) 4.98%	(0.03 ± 0.003) 0.76%
	Lba	2.7 ± 0.04 c	5.8 ± 0.04 a	4416 ± 93 c	22.1 ± 1.2 d	51.3 ± 5.3 c	26.5 ± 9.00 c	77.5 ± 18.6 d
		(114 ± 1.59) 21.1%	(425 ± 3.78) 36.6%	(29.1 ± 1.21) 32.9%	(0.21 ± 0.03) 47.5%	(0.28 ± 0.02) 27.3%	(0.04 ± 0.003) 7.55%	(0.02 ± 0.002) 1.29%
	Pep	2.5 ± 0.03 d	3.3 ± 0.03 c	4751 ± 100 b	47.6 ± 1.7 b	36.9 ± 5.1 d	25.9 ± 9.7 c	204 ± 23 c
		(116 ± 1.61) 23.2%	(271 ± 2.58) 41.0%	(51.3 ± 2.12) 54.0%	(0.38 ± 0.049) 39.9%	(0.16 ± 0.02) 21.7%	(0.02 ± 0.003) 3.86%	(0.03 ± 0.003) 0.74%
	Thy	2.9 ± 0.04 b	3.8 ± 0.04 b	4872 ± 150 b	9.5 ± 1.1 f	99.1 ± 7.1 a	31.3 ± 10.4 b	193 ± 23 c
		(133 ± 1.74) 22.9%	(229 ± 2.25) 30.1%	(30.0 ± 1.24) 30.8%	(0.06 ± 0.004) 31.6%	(0.44 ± 0.06) 22.2%	(0.04 ± 0.006) 6.39%	(0.02 ± 0.006) 0.52%
	Wne	4.4 ± 0.05 a	0.6 ± 0.04 f	5969 ± 147 a	11.4 ± 1.1 e	58.9 ± 5.8 b	29.8 ± 9.9 b	267 ± 24 b
		(58.1 ± 1.17) 6.60%	(103 ± 1.73) 85.8%	(38.8 ± 1.60) 32.5%	(0.04 ± 0.007) 17.5%	(0.22 ± 0.04) 18.7%	(0.04 ± 0.004) 6.71%	(0.05 ± 0.003) 0.94%
Malvaceae	MMa	1.4 ± 0.02 c	1.9 ± 0.02 d	3618 ± 83 c	53.1 ± 1.6 a	34.9 ± 4.4 b	31.8 ± 8.3 b	498 ± 25 a
		(24.1 ± 0.91) 8.60%	(112 ± 1.33) 29.4%	(23.5 ± 0.98) 32.5%	(0.09 ± 0.011) 8.47%	(0.05 ± 0.007) 7.16%	(0.04 ± 0.001) 6.28%	(0.04 ± 0.005) 0.40%
	Ros	1.6 ± 0.03 b	2.6 ± 0.02 c	969 ± 57 d	64.8 ± 1.6 b	29.0 ± 3.9 c	BDL	216 ± 20 b
		(167 ± 2.00) 52.2%	(304 ± 2.84) 58.4%	(14.4 ± 0.60) 74.3%	(0.91 ± 0.10) 70.2%	(0.42 ± 0.06) 72.4%	----	(0.56 ± 0.06) 13.0%
	CMa	3.5 ± 0.05 a	3.8 ± 0.04 a	6630 ± 130 a	31.2 ± 1.6 d	57.6 ± 6.5 a	23.0 ± 11.5 b	131 ± 24 c
		(261 ± 2.71) 37.3%	(347 ± 3.18) 45.6%	(79.3 ± 3.27) 59.8%	(0.40 ± 0.05) 64.1%	(0.44 ± 0.05) 38.2%	(0.04 ± 0.002) 8.70%	(0.16 ± 0.021) 6.10%
	Lin	1.4 ± 0.03 c	3.1 ± 0.03 b	4578 ± 160 b	35.8 ± 1.6 c	32.6 ± 5.6 b	46.5 ± 11.5 a	55.1 ± 24.7 d
		(33.8 ± 0.99) 12.1%	(121 ± 1.41) 19.5%	(15.8 ± 0.25) 17.2%	(0.15 ± 0.021) 20.9%	(0.08 ± 0.006) 12.2%	(0.02 ± 0.001) 2.15%	(0.04 ± 0.004) 3.63%

Average concentrations in the same column and same family, followed by a common letter, are not significantly different (*p* ≤ 0.05); F = Family; Mean values are expressed in µg g^−1^ ± standard deviation or in mg g^−1^ ± standard deviation such as Ca and K; BDL = Below the Detection Limit; Value in parentheses: concentration of each element into the infusion, expressed as mg L^−1^; all the analysis were performed in triplicate (*n* = 3); Elemental solubility expressed in %.

**Table 3 plants-11-01412-t003:** Eigenvalues of correlation matrix, and related statistics. Percentage of variance for each component (initial eigenvalues).

	1	2	3	4	5	6	7
Eigenvalue	2.16	1.27	1.07	1.05	0.66	0.54	0.26
Variance (%)	30.9	18.1	15.3	14.9	9.4	7.8	3.6
Cumulative Variance (%)	30.9	49.0	64.3	79.2	88.6	96.4	100.0

**Table 4 plants-11-01412-t004:** Correlation coefficients between attributes and PC1, PC2, and PC3.

	Components		
Attributes	PC1	PC2	PC3
Ca	0.23	0.70 *	−0.36
K	0.70 *	0.15	0.38
S	0.80 *	0.05	−0.08
Sr	0.13	0.64 **	0.596
Zn	0.85 *	−0.27	−0.23
Cu	−0.11	−0.32	0.60 **
Fe	0.48	−0.40	0.13

* Values considered strongly correlated with PC (|r| > 0.7); ** Values considered moderately correlated with PC (0.6 < |r| < 0.7); following the classification used previously.

**Table 5 plants-11-01412-t005:** Estimated daily intake (EDI) expressed in mg/kg bw/day.

Acronym/Element/Infusion Solubility (%)			EDI	RDI
Ros	Ca	52.2%	4.21 × 10^−4^	800 mg
Wne	K	85.8%	2.60 × 10^−4^	2000 mg
Art	K	63.0%	4.76 × 10^−4^	
Fen	S	96.9%	7.31 × 10^−5^	13 mg
Art	S	88.5%	7.06 × 10^−5^	
Rse	S	83.6%	4.08 × 10^−5^	
Ros	S	74.3%	3.63 × 10^−5^	
Mth	S	70.0%	4.31 × 10^−5^	
Art	Sr	89.7%	1.66 × 10^−6^	---
Ros	Sr	70.2%	2.29 × 10^−6^	
Ros	Zn	72.4%	1.06 × 10^−6^	10 mg
Mth	Cu	16.1%	2.52 × 10^−7^	1 mg
Ros	Fe	13.0%	1.41 × 10^−6^	14 mg

RDI—Recommended Daily Intake.

**Table 6 plants-11-01412-t006:** Elemental concentration present in certified standard samples (Orchard leaves -SRM 1571).

Element	Certified	Current Work	Recovery (%)
Ca	20,900 ± 300	19,700 ± 600	94
Cu	12 ± 1.0	13 ± 1.0	108
Fe	300 ± 20	278 ± 7	93
K	14,700 ± 300	14,000 ± 400	95
S	2300 *	2090	
Sr	37 *	39	
Zn	25 ± 3	23 ± 2	92

(*) Values obtained by noncertified methods.

**Table 7 plants-11-01412-t007:** Analytical conditions for ICP-AES.

Instrument	ICP Ultima
RF generator power	1.05 kw
RF frequency	40.68 MHz
Plasma gas flow rate	12 L/min
Carrier gas flow rate	1.0 L/min
Sample introduction	Miramist nebulizer
Misting chamber	Cyclonic glass chamber
Observation method	Radial
Injector tube diameter	3 mm

## Data Availability

Not applicable.

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
