# Peer review of "Elemental Composition of Commercial Herbal Tea Plants and Respective Infusions"

_plants, 2022, doi:10.3390/plants11111412_

Round 1
Reviewer 1 Report
I have browsed the manuscript entitled “Elemental composition of commercial herbal tea plants and respective infusions”. There are several questions need to be resolve.
1. The novelty aspects and its underlying concepts are not clear in this manuscript.
2. The results and discussion in the manuscript is too descriptive and lack of condensed. They are need to promote and streamline.
3.It is not clear to reader knowledge can be obtained from this article.
4.The conclusion in the manuscript is not clear, and it was lack of persuasive
Author Response
REVIEWER 1
- The novelty aspects and its underlying concepts are not clear in this manuscript.
You are not aware of the profusion of brands of herbal teas with “miraculous actions” that are offered to the public. The European and Portuguese legislation regarding these types of products is absent. Everything can be sold without any control.
I have a large experience in evaluating food supplements and I can reaffirm that some of them constitute a real danger to public health due to their high arsenic or lead contents. Thus, we implement a minor screening that must be continued, to see if herbal teas have similar problems. I can assure you that it is a novelty and what can be modified is the experimental procedure – for example, start from a 5g dry material basis, increase the infusion time to 15 minutes, or even implement a decoction process.
- The results and discussion in the manuscript is too descriptive and lack of condensed. They are need to promote and streamline.
When studying the elemental content of 25 different species it is imperative to compare our data with published data so far taking always into account the origin of the samples and the methods used. In this context, I partially agree with your remark about a too descriptive text but you must consider that this type of work is similar to others.
We have the raw material and often the indication of the origin and nothing else. We can compare the elemental composition and the origin. Of course, the data obtained is greatly influenced by the extraction method which includes the time of extraction, amount of the sample, or even a decoction process instead of infusion, but those things were addressed particularly from line 240 onwards
Nevertheless, we achieved a reduction of 56 lines in the new draft.
3.It is not clear to reader knowledge can be obtained from this article.
Similarly, to other articles, this knowledge is focused on the brands free sold in Portugal whose composition is ignored.
In this context let me point out some data published in very good journals where the number of samples is sometimes lower than ours.
- Queralt et al., (2005). X-Ray Spectrom; 34: 213–217. DOI: 10.1002/xrs.795
This work was based on 5 species and the data of infusion was obtained by ICP-AES while the data from the raw material with XRF
- Malik et al., (2008). Food Chem, 111, 520-525, doi: 10.1016/j.foodchem.2008.04.009
This work was based on 31 species and the data of infusion was obtained by ICP-OES while the data from the raw material with AAS
3) Kara, D. (2009) Food Chem. 114, 347–354, doi: 10.1016/j.foodchem.2008.09.054
This work was based on 18 species and the data of infusion was obtained by ICP-OES while the data from the raw material with ICP-MS
- Ozcan et al., (2008) Food Chem. 106, 1120–1127,https://doi.org/10.1016/j.foodchem.2007.07.042
This work was based on 20 species and the data of infusion and decoction was obtained by ICP-AES while the data from the raw material with ICP-AES
- Podwika et al., (2018) Biol Trace Elem Res. 183:389–395. DOI 10.1007/s12011-017-1140-x
This work was based on 27 species and the elemental data of the raw material (only Cu, Cd, Mn and Zn) obtained with AAS
All these studies are based on local herbal teas (sold in a particular country like Spain or Poland, but often with origins far from these countries), whose elemental composition was ignored and the aim is always to verify if there is any risk to public health derived to the consumption of these brands.
The validity of a study can not be expressed only when some harmful elements were found. If the raw material is clean regarding those elements, we can be confident in those brands and respective exporters and assume that the study is valid too.
In the Abstract we include a statement about the safety of the brands used (blue lines 30-31)
4.The conclusion in the manuscript is not clear, and it was lack of persuasive
We must stress that the conclusion was based on the selected raw material and the method of extraction using only 1 gram (dry weight) and 5 minutes of infusion. From this approach, several conclusions must be drawn such as:
- Some species are rich in particular elements
- Got samples emerge as a specie with the highest composition regarding the studied elements
- The solubility of micronutrients is poor, while the solubility of macronutrients is much higher
- The Estimated Daily Intake values which were very low, point out to a negligible contribution to the tea brew to the human daily requirements regarding essential micro and macronutrients.
- Non-essential elements (As and Pb for example), were absent in the studied samples.
Thus, the conclusions are clear but are always related to the type of experiment we used. Most probably, with longer extraction times or even applying a decoction process, the elemental transfer to the brew will be much higher
Minor Remarks
Blue lines were added – 383-385 (to fully complete the materials and methods section), based on the reference previously quoted
The authors would like to express their gratitude to both reviewers despite opinions that are sometimes divergent.
Best regards
Fernando Reboredo

Author Response
REVIEWER 2
Thank you for considering my suggestions.
The paper is very well written but some minor language and typing revision is needed.
Some minor errors were detected and corrected
It would be better to use common names or scientific names in the Abstract rather than codes, for a better understanding from those who, in search of a specific subject for example, only read the abstracts to see if the respective papers contain the information they are looking for.
Your remark was fully addressed
The subject is of great interest as herbal tea brews are largely consumed around the World and many people rely on their therapeutic properties too. So I think it would be suitable that this paper, or at least a simplified version of it, be circulated outside the scientific community too
Minor Remarks
One of the criticisms was linked with an excess of description in the text. Without alter the focus of our work and taking into account that we are dealing with 25 different species we reduced the new draft in 56 lines
In the Abstract we include a statement about the safety of the brands used (blue lines 30-31)
Blue lines (383-385) were added to complete the materials and methods section), based on the reference previously quoted
The authors would like to express their gratitude to both reviewers for their positive criticism
Best regards
Fernando Reboredo

Round 2
Reviewer 1 Report
no comments.